# Impact of Perineural Invasion and Preexisting Type 2 Diabetes on Patients with Esophageal Squamous Cell Carcinoma Receiving Neoadjuvant Chemoradiotherapy

**DOI:** 10.3390/cancers15041122

**Published:** 2023-02-09

**Authors:** Nai-Wen Su, Lai-Man Mok, Mei-Lin Chan, Hung-Chang Liu, Wei-Chin Chang, Chun-Ho Yun, Tze-Yu Shieh, Ming-Che Wu, Huan-Chau Lin, Wen-Chien Huang, Yu-Jen Chen

**Affiliations:** 1Department of Optometry, MacKay Junior College of Medicine, Nursing, and Management, Taipei 11260, Taiwan; 2Division of Hematology and Medical Oncology, Department of Medicine, MacKay Memorial Hospital, Taipei 10449, Taiwan; 3Division of Thoracic Surgery, Department of Surgery, MacKay Memorial Hospital, Taipei 10449, Taiwan; 4Department of Pathology, MacKay Memorial Hospital, New Taipei City 25160, Taiwan; 5Department of Medicine, MacKay Medical College, New Taipei City 25245, Taiwan; 6Department of Radiology, MacKay Memorial Hospital, Taipei 10449, Taiwan; 7Division of Gastroenterology, Department of Medicine, MacKay Memorial Hospital, Taipei 10449, Taiwan; 8Department of Nuclear Medicine, MacKay Memorial Hospital, Taipei 10449, Taiwan; 9Department of Radiation Oncology, MacKay Memorial Hospital, New Taipei City 25160, Taiwan; 10Department of Medical Research, China Medical University Hospital, Taichung 40402, Taiwan; 11Department of Artificial Intelligence and Medical Application, MacKay Junior College of Medicine, Nursing, and Management, Taipei 11260, Taiwan

**Keywords:** esophageal squamous cell carcinoma, perineural invasion, neoadjuvant chemoradiotherapy, type 2 diabetes

## Abstract

**Simple Summary:**

Neoadjuvant chemoradiotherapy followed by surgery is the standard treatment in locally advanced esophageal squamous cell carcinoma. This multimodality strategy provides survival benefits superior to surgery alone, especially in patients obtaining a pathological complete response (pCR). Owing to subsequent recurrence and metastasis, many patients do not achieve a pCR (non-pCR) after neoadjuvant chemoradiotherapy and therefore have very poor outcomes. It is necessary to identify poor prognostic factors. In this real-world data analysis and retrospective cohort study, we found that the presence of perineural invasion and preexisting type 2 diabetes had negative impacts on disease-free survival in the non-pCR population. Patients with a combination of both two factors had the worst survival. Our findings provide clinical information for future translational investigations and possible clinical applications.

**Abstract:**

Neoadjuvant chemoradiotherapy (neoCRT) followed by surgery is the cornerstone treatment strategy in locally advanced esophageal squamous cell carcinoma (ESCC). Despite this high- intensity multimodality therapy, most patients still experience recurrences and metastases, especially those who do not achieve a pathological complete response (pCR) after neoCRT. Here, we focused on identifying poor prognostic factors. In this retrospective cohort study; we enrolled 140 patients who completed neoCRT plus surgery treatment sequence with no interval metastasis. Overall, 45 of 140 patients (32.1%) achieved a pCR. The overall survival, disease-free survival (DFS), and metastasis-free survival was significantly better in patients with a pCR than in patients with a non-pCR. In the non-pCR subgroup, the presence of perineural invasion (PNI) and preexisting type 2 diabetes (T2DM) were two factors adversely affecting DFS. After adjusting for other factors, multivariate analysis showed that the hazard ratio (HR) was 2.354 (95% confidence interval [CI] 1.240–4.467, *p* = 0.009) for the presence of PNI and 2.368 (95% CI 1.351–4.150, *p* = 0.003) for preexisting T2DM. Patients with a combination of both factors had the worst survival. In conclusion, PNI and preexisting T2DM may adversely affect the prognosis of patients with ESCC receiving neoadjuvant chemoradiotherapy.

## 1. Introduction

Esophageal cancer (EC) is a highly aggressive disease with a grave prognosis. It is the seventh most common cancer and sixth leading cause of cancer-related mortality [1]. The two major histologic types are esophageal squamous cell carcinoma (ESCC) and esophageal adenocarcinoma, each possessing distinct geographic distribution, pathophysiology, and treatment outcomes [2]. ESCC accounts for the majority (85%) of new EC cases worldwide annually [3,4]. The highest incidence is in Eastern Asia, including Taiwan, with EC having a significant impact on health care systems [5,6]. Additionally, more than half of patients with ESCC present with locally advanced disease status. Therefore, it is important to combine treatment modalities to initially achieve local tumor control and to prevent subsequent distant spread [7].

While neoadjuvant chemotherapy is implemented in some areas [8,9,10], neoadjuvant chemoradiotherapy (neoCRT) followed by surgery is a more widely accepted treatment strategy worldwide [11,12,13]. The pivotal CROSS study, as well as other clinical trial results, demonstrated that neoCRT plus surgery provided more survival benefits than surgery alone in locally advanced ESCC [14,15]. Moreover, the benefits were maintained after long-term follow up [16,17]. Approximately 20–40% of patients with locally advanced EC achieve a pathological complete response (pCR) after neoCRT treatment and thereafter obtain superior survival outcomes [18,19,20,21,22]. In contrast, more than half of the patients have residual disease after neoCRT, resulting in future local or distant recurrences. From the literature review, patients with EC having some comorbidities, such as type 2 diabetes (T2DM) and pathological risk factors, such as perineural invasion(PNI) or vascular invasion were reported to have inferior outcomes [23,24,25,26].In this study, we collected 10-year cohort data and analyzed the survival outcomes of patients with locally advanced ESCC treated with neoCRT followed by surgery, focusing on patients not achieving a pCR (non-pCR). We found that the presence of PNI and preexisting T2DM would result in a worse survival outcome in patients.

## 2. Materials and Methods

### 2.1. Patient Eligibility

This retrospective cohort study was approved by MacKay Memorial Hospital Institutional Review Board, approval number 21MMHIS178e. Informed consent from participants was not required in this IRB-approved study. Patients with pathologically confirmed, non-metastatic, locally advanced (stage II, III, or IVa) ESCC who completed neoCRT followed by surgery were included. However, patients who died within 3 months of the surgery were excluded from the survival analysis. Cancer staging was performed according to the American Joint Committee on Cancer 8th edition for both clinical and pathological staging. The clinical staging procedures included physical examinations, panendoscopic ultrasound, chest computed tomography (CT), positron emission tomography, and bronchoscopy as needed. Feeding jejunostomy was also deployed to maintain enteric nutrition, although this was not mandatory. After surgery, patients were followed up with chest CT scans every 3–4 months during the first 2 years, and every 6 months during years 3–5. Additional imaging studies were arranged according to clinical necessity. Once the disease recurred, its management was at the discretion of the treating physician and was based on the best patient benefits.

### 2.2. Multimodality Treatments

The cases of all enrolled patients with ESCC were discussed during the multidisciplinary team (MDT) meetings before the implementation of neoCRT and before surgical resection. Radiotherapy was planned using simulation CT images. Patients were first immobilized in the supine position using an Alpha Cradle^®^ (Smithers Medical Products, Inc. North Canton, OH, USA). The CT images (Brilliance Big Bore CT simulator/Philips Medical Systems, Cleveland, OH, USA) were obtained using 3-mm-thick slices. The delineation of gross tumor volume (GTV) included the esophageal gross tumor and enlarged regional lymph nodes. The clinical target volume (CTV)-48 was defined as 0.5–1 cm outside the GTV to cover the microscopic extension. The CTV-43.2 included subclinical mucosal/submucosal disease and risky regional nodal basins such as supraclavicular, paraesophageal, paratracheal and celiac trunk regions. The planning target volume (PTV) enclosed the CTV with margins to account for possible uncertainties in patient set-up error or internal organ motion. Intensity-modulated radiation therapy or helical tomotherapy with simultaneously integrated boost technique delivered 48 Gy (2 Gy per fraction) to the PTV-48 and 43.2 Gy (1.8 Gy per fraction) to the PTV-43.2. Normal organ constraints included the following: maximum dose of spinal cord was <45 Gy, proportion of lung receiving > 20 Gy (V20) < 20% in each side lung, and mean heart dose <30 Gy. The treatment plans had to satisfy the criterion that at least 95% of the PTV was covered with the prescribed dose. RT was withheld if the patient had ≥ grade 3 decline in neutrophil or platelet counts.

For concurrent chemotherapy, we used two regimens, weekly cisplatin 20–30 mg/m^2^ plus paclitaxel 50 mg/m^2^, or weekly cisplatin 30 mg/m^2^ if self-funded paclitaxel was not feasible. Both chemotherapy regimens were administered in 5–6 cycles during RT. If patients had renal insufficiency at the beginning or during the course of chemotherapy, cisplatin was substituted with carboplatin (area under the curve 2).

After disease re-evaluation processes and operation appropriateness confirmed by MDT, surgery with esophagectomy and radical lymph node dissection was scheduled at 4–6 weeks after the end of neoCRT. In brief, the Ivor Lewis procedure with two-field lymph node dissection was selected for tumors located at the distal thoracic esophagus. The McKeown method with three-field lymph node dissection was selected for tumors located at the upper or middle thoracic esophagus. Additionally, for cervical esophageal tumors, either laryngeal-preserving surgery or pharyngo-laryngo-esophagectomy was performed depending on whether the proximal margin of the tumor was within or beyond 2 cm from the inferior margin of the pyriform sinus of the hypopharynx. Pathological response was assessed by pathologists specialized in gastroenterology (W.-C.C.). A pCR was defined as no residual tumor in the primary tumor site or in all examined lymph nodes (ypT0N0).

### 2.3. Endpoints Definition and Study Variables

All time-related endpoints were counted from the starting date of neoCRT to the date of event occurrence. The data cutoff date was 31 December 2021. Patients who were lost to follow up or those without an event until the data cutoff date were recorded as being “censored”. At least 1 year of follow up at our institute was required for living patients to be included in this study. The study endpoints were (1) overall survival (OS): death from any cause; (2) disease-free survival (DFS): any events of local or distant recurrence, including a second primary tumor, or death, depending on which occurred first; and (3) metastasis-free survival (MFS): any event of distant recurrence or death, depending on which occurred first. We analyzed clinical and pathological variables that may have prognostic impacts on survival outcomes. Clinical factors included age (<57-years vs. ≥57-years), sex, ECOG performance status (0 vs. 1/2), tumor location (cervical/upper vs. middle/lower esophagus), history of T2DM, history of head and neck cancer, chemotherapy regimens (platinum/paclitaxel vs. platinum), number of cycles delivered (<5 vs. ≥5), and time from the completion of neoCRT to operation (<6 vs. ≥6 weeks). Pathological factors comprised the pathological T stage (T0–T2 vs. T3/T4), pathological N stage (N0/N1 vs. N2/N3), cell differentiation (well/moderate vs. poor), lymphovascular invasion (LVI) and PNI (with vs. without), and dissected lymph node numbers (<15 vs. ≥15).

### 2.4. Statistical Methods and Tools

Clinical or pathological characteristic comparisons between the two groups were calculated using two-sample t-test or Fisher`s exact test. Survival curves were performed using the Kaplan–Meier method and statistical significance was examined using the log-rank test. Univariate and multivariate outcome analyses were computed using the proportional hazards method to demonstrate the survival impacts of the clinical or pathological variables. Statistical significance was set at *p* < 0.05. Statistical analyses and graphic preparation were executed using SPSS software (version 22; IBM Corp, Armonk, NY, USA).

## 3. Results

### 3.1. Patient Enrollment and Characteristics

We retrospectively collected clinicopathological information and treatment outcomes of patients with ESCC in our institute from January 2010 to December 2019. There were 216 patients with locally advanced ESCC who received neoCRT after the MDT discussion. Distant metastasis developed at the end of neoCRT in 25 of 216 patients; therefore, the interval metastasis rate was 11.6%. Finally, 154 patients underwent surgery. The detailed patient disposition process depicted in Figure 1. Overall, 140 patients were included in the survival analysis. The median age was 57-years (range, 37–81), and 130 of 140 (92.9%) patients were men. The number of patients with primary tumors at the cervical esophagus was 10 (7.1%); upper thoracic, 31 (22.1%); middle thoracic, 60 (42.9%); and lower thoracic, 39 (27.9%). The clinical stage distribution was 24 (17.1%), 89 (63.6%), and 27 (19.3%) for stages II, III and IVa. Overall, 63 (45%) patients received platinum plus paclitaxel as the concurrent chemotherapy. Of all the patients, 102 (75.6%) received five cycles of chemotherapy during irradiation. The median radiation dose was 48 Gy (range, 24–54 Gy). The median time interval from the completion of neoCRT to surgery was 41 days (range, 19–75 days). In total, 21 (15%) and 13 (9.3%) patients had a history of T2DM and head and neck cancer, respectively. Seventeen patients with T2DM had the HbA1c level checked at ESCC diagnosis. The mean (range), median and interquartile range of HbA1c level (%) were 6.62 (5.1–7.9), 6.7 and 6.1 to 7.15, respectively. Detailed patient clinical characteristics are presented in Table 1.

### 3.2. Overall Treatment Outcomes

With a median follow-up time of 59.5 months, the median OS and DFS of the overall population were 24.3 months (95% confidence interval (CI):16.6–31.9 months) and 11.2 months (95% CI: 8.4–13.9 months), respectively. Furthermore, 45 of 140 (32.1%) patients had achieved a pCR. The median OS, DFS and MFS of patients with a pCR were significantly greater than those of patients with a non-pCR (Figure 2a–c). The hazard ratios (HRs) of OS, DFS and MFS were 0.367 (95% CI: 0.226–0.594; *p* < 0.0001), 0.457 (95% CI: 0.295–0.706; *p* < 0.0001) and 0.415 (95% CI: 0.264–0.5; *p* < 0.0001), respectively, for patients achieving a pCR compared with patients with a non-pCR. The 1-year, 3-year, and 5-year OS rates were 93.3%, 65.6%, and 44% for patients with a pCR and 68.4%, 28.1%, 18.7% for patients with a non-pCR, respectively. Overall, 24 of 45 (53.3%) patients with a pCR and 78 of 95 (82.1%) patients with a non-pCR developed DFS events after neoCRT plus surgery. Separate pCR and non-pCR patient clinical characteristics are shown in Table 1.

### 3.3. Prognostic Factors in Patients with a Non-pCR

The group of patients who did not achieve a pCR after neoCRT was heterogenous, generally implying adverse survival outcomes. In total, 95 patients (67.8%) did not obtain a pCR after neoCRT. The detailed pathological features are shown in Table 2. Overall, 6 of 95 (6.3%) patients had an R1 resection; there were no R2 resection. Among the clinicopathological variables, we performed univariate analysis to define the poor prognostic factors of DFS in this patient population (Table 3). We found that patients with preexisting T2DM had worse DFS with a HR of 2.139 (95% CI 1.141–4.011, *p* = 0.018), and the median DFS was 8.36 vs. 11.21 months (log-rank test, *p* = 0.018) in patients with and without preexisting T2DM, respectively. Patients with PNI had poorer DFS with a HR of 2.449 (95% CI 1.497–4.007, *p* = 0.0001), and the median DFS was 7.84 vs. 14.75 months (log-rank test, *p* = 0.0001) in patients with and without PNI, respectively. Finally, patients with a higher pathological N stage (N2/N3 vs. N0/N1) had inferior DFS with a HR of 1.843 (95% CI 1.117–3.041, *p* = 0.017), and the median DFS was 8.49 vs. 13.41 months (log-rank test, *p* = 0.015) in patients with N2/N3 and N0/N1 disease (Figure 3a–c). There were no association among these three factors. Subsequently, we included these three variables into a multivariate analysis. After adjusting for other variables, patients with T2DM and PNI remained poor prognostic factors in this patient population. The HR was 2.354 (95% CI 1.240–4.467, *p* = 0.009) and 2.368 (95% CI 1.351–4.150, *p* = 0.003), respectively (Table 3). Focusing on these two factors, preexisting T2DM and presence of PNI, there were 61, 29 and 5 patients with neither, one, or both factors, respectively. The median DFS was 16.8 (95% CI 8.93–24.71), 10.8 (95% CI 7.80–13.78), and 5.4 (95% CI 3.59–7.23) months, and it was significantly worse in patients with both features (*p* < 0.0001) (Appendix A).

## 4. Discussion

NeoCRT followed by surgery is the cornerstone treatment for locally advanced EC [27,28]. In the current study, we reported the survival outcomes of a 10-year single institute retrospective cohort treated with this multimodality strategy. We demonstrated its feasibility in our real-world daily practice. With advancement in surgical techniques and team collaboration, approximately 7% of patients have either a cervical tumor location or clinical T4 lesion. Owing to high technical demands, these clinical features usually preclude patients from having neoCRT plus surgery as one of their curative options [29,30]. Concerning the pCR rate, our data were comparable to those reported in the literature [31,32,33,34]. While patients with a pCR did have better survival outcomes than patients with a non-pCR, the low pCR rate (20–40%) under current neoCRT regimens was a limitation. In pursuit of a higher pCR rate, immunotherapy agents have been investigated in the neoadjuvant treatment setting for EC. While most studies use combined anti-PD1/anti-PDL1 and chemotherapy without radiotherapy, the pCR results were promising [35,36,37,38,39]. However, large phase III randomized, controlled trials are awaited to provide more solid evidence in this research field.

In contrast, approximately 80% of patients with a non-pCR had subsequent DFS events in this study that brought them a very dismal prognosis. Thus, we put an emphasis on the survival prognostic factors in this patient population. After univariate and multivariate Cox regression analyses, we found that preexisting T2DM and the presence of PNI predicted poor outcomes. Patients with both factors, in particular, had a DFS result that was worse than those of patients with either one or none of the factors. T2DM and uncontrolled hyperglycemia have long been linked with worse treatment outcomes in a variety of cancer types [40,41,42,43,44]. Several studies have addressed the issue regarding T2DM and its prognostic impact in EC. All these studies were reported in a retrospective manner with heterogenous treatments, except one that used propensity score analysis. There were inconsistent results regarding T2DM being a poor prognostic factor in patients with resectable EC [45,46,47,48,49,50,51,52,53]. In the reported studies, we observed that approximately 10% of patients with EC have preexistingT2DM. Thus, T2DM and glycemic control may impact the treatment outcomes in patients with ESCC, requiring further prospective cohort investigation.

Pathological demonstration of PNI by malignant cells usually portends poor prognosis in several cancer types [54,55]. Through sophisticated laboratory works, tumor PNI tends to be a continuous, multistep process. Not only malignant cells but also neuron supporting cells (eg. Schwann cells), recruited immune cells and a network of molecules (e.g., neurotrophin, angiogenesis, and chemokines factors) are all involved in the supporting system [56,57,58]. In recent years, five PNI-centered prognostic studies on ESCC have been reported; we have summarized the results in Table 4. The prevalence of PNI ranges from 7.9% to 42.7%, but neoadjuvant treatment may decrease the PNI rate [59,60,61,62,63]. PNI tends to be a statistically significant poor indicator of OS or DFS in most reports [59,60,62]. Some studies focused on the presence of PNI being a poor prognostic factor in the pN0 subgroup patients [60,61,62]. Our data also revealed that the presence of PNI brought a significantly worse outcome (*p* = 0.003) in the node-negative subgroup (Appendix A). In contrast, it only caused a trend of inferior survival (*p* = 0.058) in the node-positive subgroup (Appendix A). Some studies disclosed that cancer patients with T2DM tended to have higher incidence of pathological PNI [64,65,66]. However, there were no association between these two poor prognostic factors in the current study. Regarding the adjuvant treatment in locally advanced EC, patients treated with neoCRT and with pathologically residual disease took nivolumab for 1 year in a previous study [67] and derived benefits from nivolumab across all prespecified subgroups. In this era of staggering medical cost, to delineate subgroups that would obtain greater benefits from adjuvant nivolumab is worthy of investigation. According to our results and the results of others, the presence of PNI may be a potential biomarker to select patients for adjuvant treatment.

There are certain limitations to this study. First, this is a retrospective cohort design and all patients were from single center which might cause bias. Thus, data interpretation and its generalizability should be cautiously considered. Second, there was no standard systemic therapy for patients with residual disease after neoCRT and individualized therapy following the MDT discussion. Third, we did not perform immunohistochemical staining with stains such as anti-S-100 or anti-CD31 to detect nerve or vascular invasion. Therefore, identifying these pathological features may be less accurate.

## 5. Conclusions

The results of this retrospective cohort study of patients with locally advanced ESCC who were treated with neoCRT followed by surgery reflected real-world practice and evidence. The initial treatment goal of achieving a pCR is paramount to endorse patients with better survival outcomes. In patients with a non-pCR, preexisting T2DM and the presence of PNI are both significant poor prognostic factors. The unfavorable impacts of these two factors and the implications for possible adjuvant treatment warrant further investigations.

## Figures and Tables

**Figure 1 cancers-15-01122-f001:**
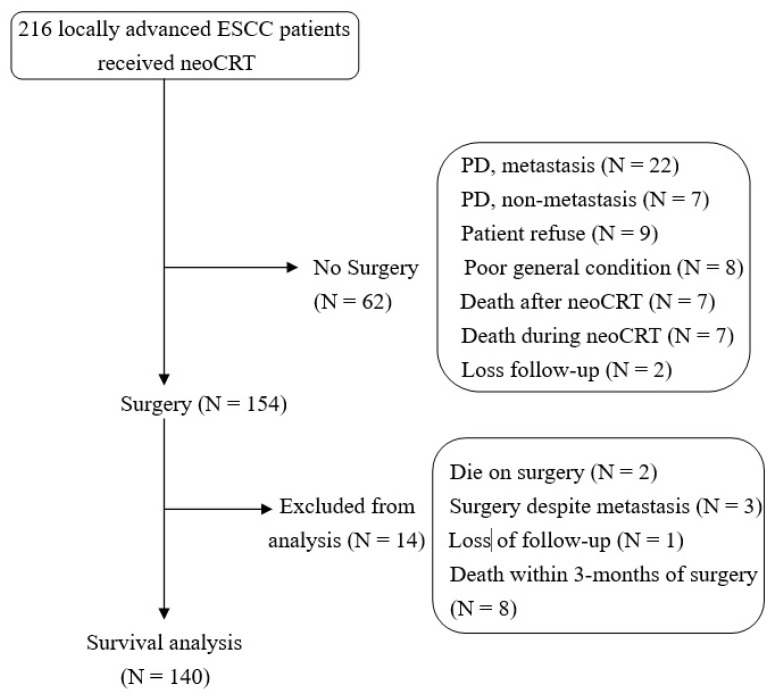
Flow chart of patient enrollment and exclusion. Abbreviations: ESCC: esophageal squamous cell carcinoma; neoCRT: neoadjuvant chemoradiotherapy; PD: progression of disease.

**Figure 2 cancers-15-01122-f002:**
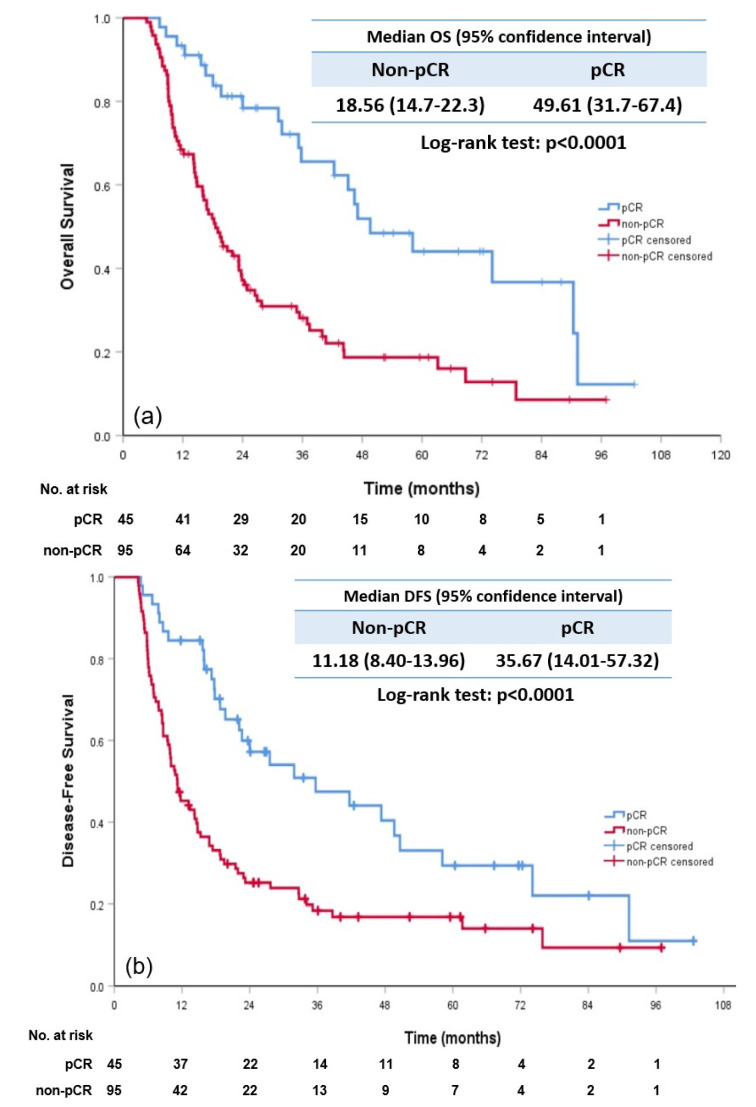
Kaplan–Meier curve for comparison between pCR and non-pCR patient subgroups. (**a**) OS; (**b**) DFS; and (**c**) MFS. Abbreviation: pCR: pathological complete response; OS: overall survival; DFS: disease-free survival; and MFS: metastasis-free survival.

**Figure 3 cancers-15-01122-f003:**
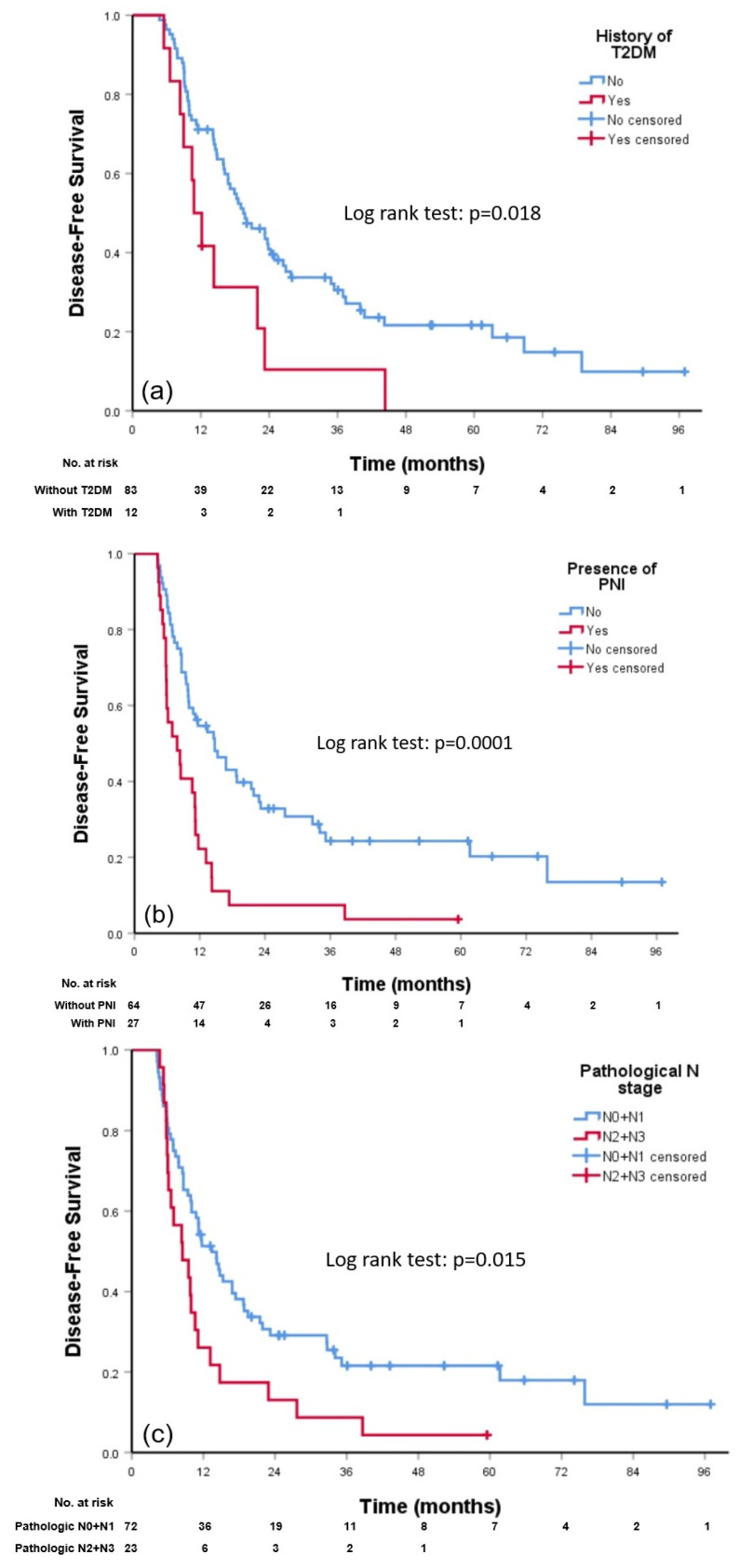
Kaplan–Meier curve of disease-free survival for comparison between patients (**a**) with vs. without preexisting T2DM; (**b**) with vs. without PNI; and (**c**) N0/N1 vs. N2/N3. Abbreviations: T2DM: type 2 diabetes; PNI: perineural invasion.

**Table 1 cancers-15-01122-t001:** Patient clinical characteristics.

Variable	Overall (N = 140)	pCR (N = 45)	Non-pCR (N = 95)
Age	57.5 ± 8.7	59.6 ± 8.8	56.6 ± 8.5
Sex Male Female	130 (92.9%) 10 (7.1%)	41 (91.1%) 4 (8.9%)	89 (93.7%) 6 (6.3%)
ECOG 0 1	42 (30%) 98 (70%)	12 (26.7%) 33 (73.3%)	30 (31.6%) 65 (68.4%)
Tumor Location Cervical Upper Thoracic Middle Thoracic Lower Thoracic	10 (7.1%) 31 (22.1%) 60 (42.9%) 39 (27.9%)	2 (4.4%) 8 (17.8%) 23 (51.1%) 12 (26.7%)	8 (8.4%) 24 (25.3%) 36 (37.9%) 27 (28.4%)
Clinical T Stage T1 T2 T3 T4	5 (3.6%) 39 (27.9%) 86 (61.4%) 10 (7.1%)	3 (6.7%) 14 (31.1%) 28 (62.2%) 0	2 (2.1%) 25 (26.3%) 58 (61.1%) 10 (10.5%)
Clinical N Stage N0 N1 N2 N3	11 (7.9%) 47 (33.6%) 63 (45%) 19 (13.6%)	2 (4.4%) 15 (33.3%) 26 (57.8%) 2 (4.4%)	9 (9.5%) 32 (33.7%) 37 (38.9%) 17 (17.9%)
Clinical Stage II III Iva	24 (17.1%) 89 (63.6%) 27 (19.3%)	7 (15.6%) 36 (80%) 2 (4.4%)	17 (17.9%) 53 (55.8%) 25 (26.3%)
CT regimen Platinum/Paclitaxel Platinum Only	63 (45%) 77 (55%)	19 (42.2%) 26 (57.8%)	44 (46.3%) 51 (53.7%)
CT Cycles ≥5 <5	102 (72.8%) 38 (27.2%)	38 (84.4%) 7 (15.6%)	70 (73.7%) 25 (26.3%)
RT Dose (cGy)	4704 ± 328.7	4742 ± 231.1	4686 ± 365.8
CRT_Op Time (days) ≤42 days >42 days	41.5 ± 10.2 78 (55.7%) 62 (44.3%)	38.8 ± 10.3 32 (71.1%) 13 (28.9%)	42.8 ± 9.9 46 (48.4%) 49 (51.6%)
History of HNSCC No Yes	127 (90.7%) 13 (9.3%)	41 (91.1%) 4 (8.9%)	86 (90.5%) 9 (9.5%)
History of T2DM No Yes	119 (85%) 21 (5%)	36 (80%) 9 (20%)	83 (87.4%) 12 (6.3%)

Abbreviations: pCR: pathological complete response; ECOG: Eastern Cooperative Oncology Group CT: chemotherapy; RT: radiotherapy; CRT: chemoradiotherapy; Op: operation; HNSCC: head and neck squamous cell carcinoma; and T2DM: type 2 diabetes.

**Table 2 cancers-15-01122-t002:** Pathological characteristics of patients with a non-pCR (N = 95).

Pathologic T Stage T0 T1 T2 T3 T4	7 (7.4%) 15 (15.8%) 35 (36.8%) 33 (34.7%) 5 (5.3%)	Differentiation Well Moderate Poor	4 (4.2%) 72 (75.8%) 19 (20.0%)
Pathologic N Stage N0 N1 N2 N3	47 (49.5%) 25 (26.3%) 18 (18.9%) 5 (5.3%)	LVI Yes No	25 (26.3%) 70 (73.7%)
Residual Disease Status Residual T Only Residual N Only Residual T + N	47 (49.5%) 7 (7.4%) 41 (43.1%)	PNI Yes No	27 (28.4%) 68 (71.6%)
ypStage Stage I Stage II Stage IIIa Stage IIIb Stage Iva	27 (28.4%) 16 (16.8%) 16 (16.8%) 26 (27.4%) 10 (10.5%)	Dissected Lymph Nodes <15 ≥15	44 (46.3%) 51 (53.7%)

Abbreviations: pCR: pathological complete response; LVI: lymphovascular invasion; and PNI: perineural invasion.

**Table 3 cancers-15-01122-t003:** Univariate and multivariate analyses of disease-free survival (DFS) in patients with a non-pCR.

	Univariate Analysis of DFS		Multivariate Analysis of DFS	
Variables	HR (95% CI)	*p*-Value	HR (95% CI)	*p*-Value
Age (years) <57 ≥57	Ref 1.081 (0.689–1.695)	0.734		
Sex Male Female	Ref 0.566 (0.178–1.800)	0.335		
ECOG 0 1	Ref 1.278 (0.787–2.075)	0.322		
Tumor Location cervical + upper middle + lower	Ref 1.321 (0.797–2.189)	0.280		
History of HNSCC No Yes	Ref 1.606 (0.768–3.360)	0.208		
History of T2DM No Yes	Ref 2.139 (1.141–4.011)	0.018	Ref 2.354 (1.270–4.467)	0.009
CT Regimen Platinum/Paclitaxel Others	Ref 1.349 (0.855–2.219)	0.199		
CT Cycles ≥5 <5	Ref 0.780 (0.472–1.289)	0.333		
neoCRT to Op time ≤42 days >42 days	- Ref 0.846 (0.542–1.320)	0.461		
Differentiation Well + moderate Poor	Ref 1.434 (0.845–2.434)	0.182		
Dissected LNs ≥15 <15	Ref 0.922 (0.587–1.447)	0.723		
Presence of LVI No Yes	Ref 1.235 (0.751–2.031)	0.405		
Presence of PNI No Yes	Ref 2.449 (1.497–4.007)	0.001	Ref 2.368 (1.351–4.150)	0.003
Pathological T Stage T0–T2 T3–T4	Ref 1.390 (0.887–2.177)	0.151		
Pathological N Stage N0–N1 N2–N3	Ref 1.843 (1.117–3.041)	0.017	Ref 1.231 (0.688–2.203)	0.483
Presence of LVI No Yes	Ref 1.235 (0.751–2.031)	0.405		
Presence of PNI No Yes	Ref 2.449 (1.497–4.007)	0.001	Ref 2.368 (1.351–4.150)	0.003
Pathological T Stage T0–T2 T3–T4	Ref 1.390 (0.887–2.177)	0.151		

Abbreviations: HR: hazard ratio; CI: confidence interval; Ref: reference for comparison; HNSCC: head and neck squamous cell carcinoma; T2DM: type 2 diabetes; CT: chemotherapy; neoCRT: neoadjuvant chemoradiotherapy; Op: operation; LN: lymph node; LVI: lymphovascular invasion; PNI: perineural invasion.

**Table 4 cancers-15-01122-t004:** Summary of PNI-centered ESCC studies in the literature.

Author/Reference /Country	Year of Study	Total Patient Number	Pre-Op Treatment	PNI No. (Percentage)	Results
Zhou, et al. [59] China	2017–2020	321	neoCRT (all patients)	57 (17.8%)	Inferior OS, DFS (univariate only)
Chen, et al. [60] China	2000–2007	433	No pre-op Treatment	209 (47.7%)	Inferior OS in overall and pN0 subgroup
Gou, et al. [61] China	2009–2013	162 (all pN0M0)	No pre-op Treatment	119 (73.5%)	Inferior OS in pN0 patients
Kim, et al. [62] Korea	2007–2016	316	Neoadjuvant treatment (22.2%)	25 (7.9%)	Inferior DFS in overall and pN0 subgroup
Zhang, et al. [63] China	2017–2018	794	No pre-op Treatment	125 (15.7%)	Not a poor prognostic factor

Abbreviation: PNI: perineural invasion; ESCC: esophageal squamous cell carcinoma; OS: overall survival; DFS: disease-free survival; pN0: pathological node negative.

## Data Availability

Deidentified data of the study are available and will be provided by reasonable request to the corresponding author.

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
