# Peer review of "Impact of Perineural Invasion and Preexisting Type 2 Diabetes on Patients with Esophageal Squamous Cell Carcinoma Receiving Neoadjuvant Chemoradiotherapy"

_cancers, 2023, doi:10.3390/cancers15041122_

Round 1

Reviewer 1 Report

 This study is a retrospective analysis using clinical and pathological factors to identify prognostic factors in ESCC patients received neoadjuvant CCRT followed by surgery without pCR. In general, this investigation focused on a practical issue in treatment of ESCC with clinical relevance. Several issues need to be improved. Minor revision is suggested.

Specific comments:

1.      In Abstract, the meaning of sentence “. Patients a combination of both factors had the worst survival.” is unclear.

2.      In Introduction, the description of literature survey for T2DM could be added.

3.      A format error in Table 1 is noted.

4.      In Discussion, the possible relationship between PNI and T2DM needs to be addressed.

5.      In latest NCCN guideline, the pre-OP assessment is recommended at 5–8 weeks after completion of neoadjuvant CCRT. In this study, the OP was scheduled at 4–6 weeks after neoadjuvant neoadjuvant CCRT. The authors could elaborate this different time arrangement.

Reviewer 2 Report

Authors showed that patients with PNI and T2DM had a negative impact in the non-pCR population after treated with NeoCRT followed by surgery. It is well known fact that patients with pCR have a better prognosis than non-pCR. Therefore, it is very interesting to know what kind of factors would help improve the pCR rate. However, there are several concerns in this study.

1. It seems that there are several differences in patients' background, such as tumor location, cT and cN. Difference in these factors are usually afffect the prognosis and might affect the impact after NeoCRT. Therefore, it might be better using a propensity score match to uniform the patients' background.

2. Authors mentioned that  glycemic control may impact the treatment. Do you think this result would have changed if the glycemic control was good? Also, please provide the HbA1c or glycoalbumin level or something similar for the patients with T2DM. This would help readers to know whether patients had good or bad glycemic control. 

Round 2

Reviewer 2 Report

Authors answered my comments. I recommend this article to be published.

Author Response

Comments and Suggestions for Authors

Authors answered my comments. I recommend this article to be published.

Response: Thanks to the reviewers.
